Differences in and associations between belief in just deserts and human rights restrictions over a 3-year period in five countries during the COVID-19 pandemic

Murakami Michio michio@cider.osaka-u.ac.jp 1
Hiraishi Kai 2
Yamagata Mei 1 3 4
Nakanishi Daisuke 5
Ortolani Andrea 6 7
Mifune Nobuhiro 8
Li Yang 9
Miura Asako 1 3
1 Center for Infectious Disease Education and Research, Osaka University , Suita , Osaka , Japan
2 Faculty of Letters, Keio University , Minato-ku , Tokyo , Japan
3 Graduate School of Human Sciences, Osaka University , Suita , Osaka , Japan
4 Faculty of Culture and Information Science, Doshisha University , Kyotanabe , Kyoto (Current Address) , Japan
5 The Faculty of Health Sciences, Hiroshima Shudo University , Hiroshima , Hiroshima , Japan
6 Faculty of Law and Political Science, Rikkyo University , Toshima-ku , Tokyo , Japan
7 Institute of Humanities and Social Sciences, University of Tsukuba , Tsukuba , Ibaraki (Current Address) , Japan
8 School of Economics & Management, Kochi University of Technology , Kochi , Kochi , Japan
9 Graduate School of Informatics, Nagoya University , Nagoya , Aichi , Japan
Chen Chong
Electronic publication date: 2023 Sep 28
Publication date: 2023
Volume: 11
Electronic Location ID: e16147
Received 2023 Mar 31; Accepted 2023 Aug 30
Copyright: ©2023 Murakami et al.
Copyright year: 2023
Copyright holder: Murakami et al.
License: This is an open access article distributed under the terms of the Creative Commons Attribution License, which permits unrestricted use, distribution, reproduction and adaptation in any medium and for any purpose provided that it is properly attributed. For attribution, the original author(s), title, publication source (PeerJ) and either DOI or URL of the article must be cited.
License URL: https://creativecommons.org/licenses/by/4.0/

Keywords: Belief in just deserts, Belief in a just world, COVID-19, Cross-country comparison, Human rights restrictions, Prejudice

Funding: JSPS KAKENHI 19H01750 “The Nippon Foundation - Osaka University Project for Infectious Disease Prevention” This work was supported by JSPS KAKENHI Grant Number 19H01750 and “The Nippon Foundation - Osaka University Project for Infectious Disease Prevention.” The funders had no role in study design, data collection and analysis, decision to publish, or preparation of the manuscript.

==============================
Discrimination, which arose during the coronavirus disease 2019 outbreak, is a global public health issue. This study aimed to provide fundamental knowledge in proposing control measures to mitigate discrimination. We focused on two psychological variables: belief in just deserts (BJD, i.e., the belief that the infected individual deserves to be infected), a psychological factor that potentially promotes discrimination and prejudice, and human rights restrictions (HRR; i.e., the degree of individuals’ agreement with government restrictions on citizens’ behavior during emergencies). Differences in these items, as well as their annual trends from 2020 to 2022, were examined in Japan, the United States (US), the United Kingdom (UK), Italy, and China. In addition, the associations between BJD and HRR by country and year and the direction of the associations between them in Japan and Italy were analyzed. Online surveys were conducted annually, with 392–518 participants per country and year. The BJD was higher in Japan and lower in the UK. BJD increased significantly from 2020 to 2021 in all countries, except in China. Meanwhile, HRR was higher in China and lower in Japan. The HRR decreased from 2020 to 2021 in Japan and decreased from 2020 to 2022 in the US, the UK, and Italy. There were significant positive associations between BJD and HRR in Japan and Italy. Cross-lagged panel models revealed positive bidirectional associations between BJD and HRR in Japan and Italy, respectively, indicating that the HRR declined among those with weak BJD and that the BJD increased among those with high HRR. In Japan and Italy, the dissemination of public messages targeting those with a high HRR in the early stages of an infectious disease outbreak could potentially mitigate the adverse impact of the BJD, eventually reducing discrimination, especially when the infection is not attributed to the fault of the infected individuals.

Introduction

Discrimination and prejudice against infectious diseases are global public health issues. Historically, outbreaks of infectious diseases have led to discrimination and prejudice. Even when the mechanisms of infection are scientifically unknown, isolation from infected individuals is considered effective. Exclusionary control measures are often taken not only against infected individuals, but also against out-groups that are different from themselves. The plague pandemic of the 14th century led to the discrimination and persecution of Jews (Cantor, 2001). In recent years, there has been repeated discrimination and prejudice against infected individuals and certain groups regarding acquired immunodeficiency syndrome, Zika virus, leprosy, and other diseases (Baldassarre et al., 2020; Rahman et al., 2022; Stuber, Meyer & Link, 2008).

Discrimination and prejudice against specific races and occupations have occurred worldwide since the early stages of the coronavirus disease 2019 (COVID-19) outbreak (Bhanot et al., 2021; Devakumar et al., 2020; Lu et al., 2021). In addition, “self-restraint police” or “groups of local vigilantes” demanded strong infection control measures from others in several countries, including Japan and the United States (US) (Denyer & Kashiwagi, 2020; Ortiz, 2020). One psychological factor that potentially promotes such discrimination and prejudice is the belief in just deserts (BJD; i.e., the belief that the infected individual deserves to be infected) (Miura, Hiraishi & Nakanishi, 2020; Murakami et al., 2022). The BJD is based on belief in immanent justice (BIJ; i.e., the tendency to perceive or see justice in the events that have occurred) (Maes, 1998), which is one of the subscales of belief in a just world (i.e., the belief that individuals live in a just world where they get what they deserve and deserve what they get) (Lerner, 1980).

People who believe in a just world report higher levels of various prejudices such as race, fatness, depression, and cancer patients (Crandall & Eshleman, 2003). After the COVID-19 outbreak, studies reported a preliminary result that BJD in Japan was higher than that in the US, the United Kingdom (UK), Italy, and China in March 2020 (Miura, Hiraishi & Nakanishi, 2020). Further, there was a positive association between BJD and human rights restrictions (HRR; i.e., the degree of individuals’ agreement with government restrictions on citizens’ behavior during emergencies) in August 2020 in Japan (Murakami et al., 2022). However, although it has been noted that sociopsychological factors related to infectious diseases, such as risk perception of COVID-19 and adherence to infection control measures, vary among countries (Dryhurst et al., 2020; Van Bavel et al., 2022), it is unclear how BJD and HRR vary among countries and years.

Furthermore, one interpretation of the association between BJD and HRR in Japan (Murakami et al., 2022) is that those with a high BJD tend to seek strong restrictions against others, just as those with high BIJ tend to consider that some harm is the victim’s fault or to punish the victim (Murayama & Miura, 2015). However, it is unclear whether the association between BJD and HRR would hold true in countries other than Japan or for a certain period of time after 2020. The direction of the association between the BJD and HRR remains unclear. Clarifying this would provide evidence for proposing control measures to mitigate discrimination and prejudice and resolve conflicts among different cultures. However, to the best of our knowledge, no such study has been conducted thus far.

Therefore, this study firstly aimed to investigate the differences in BJD and HRR among countries and years. Secondly, the associations between BJD and HRR in each country and year were analyzed in an exploratory manner. Finally, the direction of the associations between BJD and HRR in countries where positive associations between BJD and HRR were identified was examined. Toward this goal, a 3-year survey was conducted in five countries (Japan, the US, the UK, Italy, and China), and a cross-lagged panel model was used to examine the direction of the associations.

Methods

Ethics

This study was approved by the Osaka University Graduate School of Human Sciences Research Ethics Committee (approval no. HB022-007). Participant consent was obtained before starting the questionnaire. Only participants who chose to consent to the survey online were included in the study. This study was registered at https://osf.io/9cbhr; https://osf.io/gdrpj; 2021 https://osf.io/fx274; and 2022 https://osf.io/3vycq.

Control measures and number of COVID-19 reported cases in each country

The World Health Organization considered COVID-19 a Public Health Emergency of International Concern on January 30, 2020, and declared it a pandemic on March 11, 2020 (Cucinotta & Vanelli, 2020). Lockdowns and similar control measures have been implemented in various countries, but the policies, penalties, and legal bases for such control measures have differed. According to Otsuyama et al. (2020), these control measures can be divided into four categories: (1) protection measures/micro-management (path identification and containment), (2) top-down authority (control by legal penalties), (3) state based (regulation by states), and (4) a voluntary refrain approach (control based on request for self-restraint). Measures in Japan are classified as a voluntary refrain type; those in the US, state based type; those in the UK and Italy, top-down authority types; and those in China, a combination of a protection-measures/micro-management type and a top-down authority type.

The cumulative number of COVID-19 reported cases per million people as of March 27, 2020, March 15, 2021, and March 14, 2022 (i.e., the start date of the survey for each year) was 12, 3619, and 46896 in Japan; 311, 87519, and 235557 in the US; 356, 63155, and 291827 in the UK; 1465, 54853, and 227024 in Italy; and 57, 69, and 98 in China, respectively (Mathieu et al., 2020). It should be noted that this is the number of reported cases and may differ from the total number of infections.

Study design and participants

In this longitudinal study, participants aged ≥18 years from Japan, the US, the UK, Italy, and China were recruited for an online survey. Participants were recruited via CrowdWorks in Japan, Prolific.co in the US and UK, Clickworker and Prolific.co in Italy, and TenCent in China. The participants received the following amounts for compensation in the study participation: 130 JPY in Japan; 1.25 GBP in the US, UK, and Italy; and 8 RMB in China. The first survey was conducted on March 27–28, 2020 in Japan, the US, and the UK; March 27–29, 2020 in Italy; and April 21–25, 2020 in China. The second survey was conducted on participants who participated in the first survey and were supplemented with new monitors. The second survey was conducted on March 15–19, 2021 in Japan, the UK, and China and on March 15–20, 2021 in the US and Italy.

The third survey included the same sample of participants who participated in the second survey, as well as new participants in Japan, the US, the UK, and Italy. In China, data were collected from participants who participated in either the first or second survey or new participants. The third survey was conducted from March 14 to 18, 2022 in Japan, the UK, and Italy; from March 14 to 19, 2022 in the US; and from March 14 to 23, 2022 in China. The target sample size was set so that approximately 400 valid samples were collected for each survey in each country. Inattentive participants were identified and excluded from the study using an instruction manipulation check (IMC) and a directed questions scale (DQS). The concepts of IMC and DQS were proposed in Oppenheimer, Meyvis & Davidenko (2009) and Maniaci & Rogge (2014), respectively. Both IMC and DQS are publicly available and their combination can be used to identify inattentive participants (Miura & Kobayashi, 2019). IMC and DQS questionnaires created by the authors were used in this study. All the questionnaires in Japanese, English, Italian, and Chinese are included in the Supplementary File.

The IMC instructed participants to select the bottom arrow instead of selecting “Start Answer” at the beginning of the survey. Participants who did not follow the first set of instructions were presented with a warning letter, and those who violated the instructions the second time were excluded from the survey. The DQS instructed participants to choose one of seven options. The participants underwent DQS three times during the course of the survey. Participants who violated the instructions three times were excluded. We also excluded 11 and 2 participants obtained from the same e-mail addresses in the 2021 and 2022 surveys in China, respectively. Furthermore, those whose gender was inconsistent across multiple surveys were also excluded. In 2020, 2021, and 2022, 4, 10, and 8 participants in Japan; 1, 2, and 2 in the US; 2, 3, and 3 in the UK; 2, 7, and 7 in Italy; and 0, 2, and 2 participants in China were excluded.

Table 1 Sample size and information on age, gender, academic career, children under junior high school age, and elderly people over 65.

	Year	Japan	The United States	The United Kingdom	Italy	China	
		Whole the participants	Men and Women only	Whole the participants	Men and Women only	Whole the participants	Men and Women only	Whole the participants	Men and Women only	Whole the participants	Men and Women only	
Size of full samples (including inattentive respondents) in each year	2020	525	–	446	–	443	–	536	–	902	–	
2021	446	–	439	–	426	–	423	–	649	–	
2022	443	–	432	–	419	–	420	–	672	–	
Size of valid samples in each year	2020	400	396	399	394	402	401	476	473	518	510	
2021	392	390	399	392	399	394	394	392	421	415	
2022	393	387	398	389	397	393	394	387	432	423	
Size of valid samples in each year	2020 only	241	239	247	242	168	168	411	408	478	470	
2021 only	81	81	195	189	100	96	150	150	361	355	
2022 only	142	138	275	267	194	190	164	159	397	388	
2020 and 2021	60	60	81	81	96	95	14	14	28	28	
2020 and 2022	0	0	0	0	0	0	0	0	3	3	
2021 and 2022	152	152	52	51	65	65	179	177	23	23	
2020, 2021, and 2022	99	97	71	71	138	138	51	51	9	9	
Size of unique valid samples	2020–2022	775	767	921	901	761	752	969	959	1299	1276	
Age is presented as the mean (SD).	2020	35.9 (10.3)	35.9 (10.3)	36.2 (12.7)	36.4 (12.8)	35.5 (12.6)	35.5 (12.7)	34.1 (11.5)a	34.1 (11.5)	23.3 (6.4)	23.2 (6.3)	
2021	39.6 (10.1)	39.6 (10.1)	36.1 (13.2)	36.2 (13.2)	39.2 (12.9)	39.4 (12.9)	26.7 (7.4)	26.7 (7.4)	21.9 (4.0)	21.9 (4.0)	
2022	41.4 (10.6)	41.5 (10.6)	37.5 (14.1)	37.7 (14.2)	41.2 (12.9)	41.3 (12.9)	28.3 (8.0)	28.4 (8.0)	24.2 (6.8)	24.2 (6.7)	
Gender [Man, Woman, Don’t answer (n)]	2020	117, 279, 4	117, 279	182, 212, 5	182, 212	116, 285, 1	116, 285	225, 248, 1a	225, 248	353, 157, 8	353, 157	
2021	161, 229, 2	161, 229	175, 217, 7	175, 217	104, 290, 5	104, 290	227, 165, 2	227, 165	88, 327, 6	88, 327	
2022	160, 227, 6	160, 227	115, 274, 9	115, 274	87, 306, 4	87, 306	201, 186, 7	201, 186	143, 280, 9	143, 280	
Academic career [university degree or higher, less than a university degree (n)]	2020	199, 201	197, 199	246, 153	242, 152	230, 172	229, 172	294, 180a	293, 180	355, 163	351, 159	
2021	215, 177	215, 175	255, 144	251, 141	235, 164	231, 163	281, 113	279, 113	352, 69	347, 68	
2022	219, 174	216, 171	227, 171	225, 164	211, 186	208, 185	305, 89	298, 89	370, 62	363, 60	
Children under junior high school age in the family [presence, absence (n)]	2020	124, 276	124, 272	97, 302	97, 297	113, 289	113, 288	112, 362a	112, 361	222, 296	220, 290	
2021	116, 276	116, 274	98, 301	96, 296	134, 265	134, 260	38, 356	38, 354	171, 250	168, 247	
2022	107, 286	107, 280	84, 314	83, 306	139, 258	139, 254	28, 366	28, 359	144, 288	143, 280	
Elderly people over 65 in the family [presence, absence (n)]	2020	76, 324	73, 323	66, 333	66, 328	61, 341	61, 340	111, 363a	111, 362	190, 328	187, 323	
2021	110, 282	108, 282	61, 338	60, 332	57, 342	55, 339	79, 315	78, 314	145, 276	143, 272	
2022	117, 276	115, 272	73, 325	73, 316	60, 337	60, 333	83, 311	81, 306	111, 321	109, 314	
Notes.

SD standard deviation

n number

a Two individuals had missing data.

The size of the full sample in each country and year ranged from 419 to 902, yielding 392–518 valid samples (Table 1). The proportion of those excluded from the full sample (exclusion rate) ranged from 5.3% to 42.6% (mean, 15.5%). The number of valid unique samples for 2020–2022 was 775 in Japan, 921 in the US, 761 in the UK, 969 in Italy, and 1299 in China. The number of valid samples participating from 2020 to 2022 was 99 in Japan, 71 in the US, 138 in the UK, 51 in Italy, and nine in China. Crude results in 2020 for the item regarding “I think anyone who gets infected with the Coronavirus (COVID-19) got what they deserved” have already been reported in a previous report (Miura, Hiraishi & Nakanishi, 2020).

Survey items

BJD was assessed using the following two items (Murakami et al., 2022): “If anyone had been infected with the Coronavirus (COVID-19), I think it was their fault” and “I think anyone who gets infected with the Coronavirus (COVID-19) got what they deserved.” Responses were provided using a 6-point Likert scale (i.e., “strongly disagree (1)” to “strongly agree (6)”). As previously described (Murakami et al., 2022), the BJD questionnaire was developed based on the BIJ concept in consultation with multiple researchers with expertise in the BIJ. The questionnaire was prepared in Japanese, English, Italian, and Chinese and checked by one or more native speakers. Furthermore, back-translation to Japanese was performed to confirm the accuracy of the translated questionnaire.

HRR was evaluated using the following six items (Murakami et al., 2022): (1) “In emergencies, it is better to follow government requests for restrictions on freedom of movement,” (2) “In emergencies, it is better to follow government requests for restrictions on freedom of speech,” (3) “In emergencies, anyone who goes out against government lockdown policy should be punished by law,” (4) “In emergencies, speech contrary to government policy should be punished by law,” (5) “In emergencies, every citizen should watch over to ensure that government policies are respected,” and (6) “In emergencies, every citizen can autonomously take action to ensure that government policies are respected.” Responses were also provided with a 7-point Likert scale (i.e., “strongly disagree (1)” to “strongly agree (7)”). As with the BJD, the questionnaire was prepared in Japanese, English, Italian, and Chinese and checked by one or more native speakers. Back-translation to Japanese was also performed to confirm the accuracy of the translated questionnaire.

In addition, the participants were enquired about their age, gender (man, woman, or don’t answer), academic career, presence or absence of children under junior high school age, and existence of elderly people over 65 in the family. Academic career was classified into two categories: university degree or higher and less than a university degree. All the items in Japanese, English, Italian, and Chinese as well as raw data, are included in the Supplementary File. Other variables, such as the pathogen and moral domains of the Three Domains Disgust Sensitivity Scale (Tybur, 2009), were also included in the questionnaire. However, these variables were not used because they were outside the scope of this study. Details of the questionnaires are available elsewhere: https://osf.io/9cbhr, https://osf.io/gdrpj, 2021 https://osf.io/fx274, and 2022 https://osf.io/3vycq.

Statistical analysis

First, we checked the reliability of BJD and HRR. The Spearman-Brown coefficient for BJD in all data was 0.793, and the Cronbach’s α for HRR was 0.835. The values for each year for each country are presented in Tables S1 and S2. As the values were sufficiently high, the mean values for the two items of the BJD and the six items of the HRR were used.

Table 2 Belief in just deserts by country and year.

Data are shown as the mean (95% confidence interval) [standard deviation]. Simple main effects are adjusted by Bonferroni correction: P values are multiplied by the number of groups (i.e., 5 for countries and 3 for years). Interaction: P < 0.001, partial η2 = 0.006.

	Japan	The United States	The United Kingdom	Italy	China	
2020	2.27 (2.19–2.36) [1.05]a;Y	1.50 (1.41–1.58) [0.68]c;Y	1.41 (1.33–1.50) [0.66]c;Y	1.67 (1.59–1.74) [0.83]b;Y	1.77 (1.70–1.85) [0.85]b;X	
2021	2.50 (2.41–2.58) [1.08]a;X	1.93 (1.85–2.02) [0.92]b;X	1.60 (1.51–1.68) [0.84]c;X	1.94 (1.86–2.03) [0.83]b;X	1.76 (1.67–1.84) [0.81]c;X	
2022	2.52 (2.44–2.61) [1.07]a;X	1.81 (1.73–1.90) [0.80]b;X	1.54 (1.46–1.63) [0.73]c;X,Y	1.81 (1.73–1.90) [0.85]b;X	1.84 (1.76–1.92) [0.89]b;X	
Notes.

a–c Different letters represent significant differences (P < 0.05) among countries as a simple main effect.

X–Y Different letters represent significant differences (P < 0.05) between years as a simple main effect.

Two-way multivariate analysis of variance (MANOVA) was used to analyze the differences in BJD and HRR among the five countries and three years. Simple main effects were adjusted by Bonferroni correction, that is, P values were multiplied by the number of groups (i.e., five for countries and three for years). To conduct sensitivity analysis, a two-way multivariate analysis of covariance (MANCOVA) was performed with age, gender, academic career, and presence or absence of children and elderly people as covariates (participants with “don’t answer” or missing data for gender were excluded for the adjustment). Subsequently, the same approach was used for a two-way MANOVA and two-way MANCOVA with age, gender, academic career, and presence or absence of children and elderly people as covariates in the evaluation of only first-time participants.

Pearson correlations were calculated for each year in each country to confirm the association between BJD and HRR. Bonferroni correction was applied to the analysis. P values were multiplied by the number of stratified analyses (i.e., 15). To conduct sensitivity analysis, partial correlation analysis was performed while controlling for age, gender, academic career, and presence or absence of children and elderly people. The 95% confidence intervals (CI) for the partial correlation coefficients were calculated using the bootstrap method (1,000 samples). As with Pearson correlations, P values were adjusted by Bonferroni correction.

In Japan and Italy, where significant positive associations were found between BJD and HRR, we examined the directionality of the association between the two items among participants who completed all three surveys (men or women). The sample sizes were 97 and 51 in Japan and Italy, respectively. First, because the participants participated in all three surveys and were different from the participants of the analysis described above, a two-way MANOVA (mixed design) was used to check for differences in BJD and HRR between countries and among years. The Pearson correlation coefficients between BJD and HRR in each country for each year were also calculated. Bonferroni correction was also applied in the analysis, that is, P values were multiplied by the number of stratified analyses (i.e., 6). The directionality of the association between BJD and HRR was then examined through simultaneous analysis of multiple populations using a cross-lagged panel model incorporating age in 2020 and gender as covariates. Academic career and presence or absence of children and elderly people were not included as covariates because they did not improve the model fitting: chi-square (CMIN) < 0.001, comparative fit index (CFI) = 0.823, standardized root mean squared residual (SRMR) = 0.078, root mean square error of approximation (RMSEA) = 0.077, and Akaike’s information criterion = 255.988. The bias-corrected 95% CI was calculated using the bootstrap method (1,000 samples). All statistical analyses were performed using SPSS and AMOS 28 (IBM, Chicago, IL, U.S.). Statistical significance was set at P < 0.05.

Results

Differences in BJD and HRR by countries and years

The interaction term for the BJD in each country and year was significant (partial η2 = 0.006, Table 2). In all years, Japan had a significantly higher BJD than the other countries. The UK had a significantly lower BJD than the other countries, except for the US in 2020 and China in 2021. Except for China, all countries had a significant increase in BJD from 2020 to 2021. There was no significant difference in the BJD between 2021 and 2022. The HRR for each country and year are listed in Table 3. The interaction term was significant (partial η2 = 0.022). In all years, Japan had a significantly lower HRR than other countries, except the US in 2022, whereas China has a significantly higher HRR than other countries. Except for China, all countries showed significantly lower HRR in 2021 than in 2020. In Japan, there was no significant difference in the HRR between 2021 and 2022; in the US, the UK, and Italy, the HRR further significantly decreased from 2021 to 2022. Similar results were obtained for sensitivity analyses, including MANCOVA controlled for age, gender, academic career, and presence or absence of children and elderly people (Tables S3 and S4), MANOVA for first-time participants only (Tables S5 and S6), and MANCOVA controlled for age, gender, academic career, and presence or absence of children and elderly people for first-time participants only (Tables S7 and S8).

Table 3 Human rights restriction by country and year.

Data are shown as the mean (95% confidence interval) [standard deviation]. Simple main effects are adjusted by Bonferroni correction: P values are multiplied by the number of groups (i.e., 5 for countries and 3 for years). Interaction: P < 0.001, partial η2 = 0.022.

	Japan	The United States	The United Kingdom	Italy	China	
2020	3.71 (3.61–3.81) [0.94]e;X	4.07 (3.97–4.17) [1.01]d;X	4.95 (4.85–5.05) [0.96]b;X	4.47 (4.38–4.56) [1.11]c;X	5.95 (5.86–6.03) [0.94]a;X	
2021	3.53 (3.43–3.63) [0.96]d;Y	3.87 (3.77–3.97) [1.12]c;Y	4.08 (3.98–4.18) [1.13]b;Y	3.98 (3.88–4.08) [0.96]b,c;Y	5.82 (5.73–5.92) [0.95]a;X	
2022	3.37 (3.27–3.47) [0.98]d;Y	3.53 (3.43–3.63) [1.11]c,d;Z	3.86 (3.76–3.96) [1.07]b;Z	3.73 (3.63–3.83) [0.99]b,c;Z	5.82 (5.72–5.92) [0.97]a;X	
Notes.

a–c Different letters represent significant differences (P < 0.05) among countries as a simple main effect.

X–Y Different letters represent significant differences (P < 0.05) between years as a simple main effect.

Association between BJD and HRR in each year in each country

Table 4 shows the association between the BJD and HRR in each country for each year. In all years, there were significant positive associations between BJD and HRR in both Japan and Italy. The Pearson correlation coefficients were 0.239 (95% CI [0.145–0.330]), 0.260 (0.165–0.350), and 0.464 (0.383–0.538) for 2020, 2021, and 2022 in Japan and were 0.175 (0.087–0.261), 0.211 (0.115–0.304), and 0.262 (0.167–0.352) in Italy, with an increasing trend from 2020 to 2022 in both countries. In contrast, no significant correlations were found for the other countries. These results were similar to those of the partial correlations analysis after controlling for age, gender, academic career, and presence or absence of children and elderly people (Table S9).

Table 4 Pearson correlation between belief in just deserts and human rights restriction by country and year.

P values are adjusted by Bonferroni correction, that is, P values are multiplied by the number of stratified analyses (n = 15).

	Japan	The United States	The United Kingdom	Italy	China	
	r (95% CI)	P	r (95% CI)	P	r (95% CI)	P	r (95% CI)	P	r (95% CI)	P	
2020	0.239 (0.145–0.330)	<0.001	0.066 (−0.032–0.163)	1.000	−0.034 (−0.131–0.064)	1.000	0.175 (0.087–0.261)	0.002	−0.124 (−0.208–−0.038)	0.072	
2021	0.260 (0.165–0.350)	<0.001	0.117 (0.019–0.212)	0.295	0.133 (0.036–0.228)	0.115	0.211 (0.115–0.304)	<0.001	−0.093 (−0.187–0.003)	0.852	
2022	0.464 (0.383–0.538)	<0.001	0.144 (0.046–0.239)	0.061	0.039 (−0.060–0.137)	1.000	0.262 (0.167–0.352)	<0.001	−0.039 (−0.133–0.055)	1.000	
Notes.

CI confidence interval

Directionality of the associations between BJD and HRR using the cross-lagged panel model

Tables S10 and 11 show the BJD and HRR for each year in Japan and Italy, respectively, for those who participated in the three surveys. Interaction terms were not significant for either BJD or HRR (partial η2 = 0.008 for BJD and 0.017 for HRR). For BJD, country differences were significant but year differences were not. For the HRR, the country and year differences were both significant. However, these results had a weak statistical power owing to the small sample size. Overall, the trends in BJD and HRR were similar to those observed in the analyses of all participants (Tables 2 and 3).

The associations between the BJD and HRR for each year and country are shown in Table S12. In Japan, positive associations were found between 2021 and 2022. In Italy, this association was weak in 2020 and strong in 2022. The Pearson correlation coefficient for 2022 in Italy tended to be particularly higher compared to that analyzed for all participants (Table 4).

The cross-lagged panel model was found to have a good fit based on the following results (Fig. 1): chi-square (CMIN) < 0.001, CFI = 0.887, SRMR = 0.069, and RMSEA = 0.088. Between 2021 and 2022, positive associations were found between BJD and HRR and between HRR and BJD in Japan. A similar trend was observed in Italy, although the statistical power was particularly weak because the sample size was 51. There was no large difference in structure between Japan and Italy, except for the path from the BJD in 2021 to the BJD in 2022 and the partial correlation between the BJD and HRR in 2022.

Figure 1 Relationship between belief in just deserts (BJD) and human rights restriction (HRR) in Japan and Italy (cross-lagged panel model).

The values represent standardized estimates (bias-corrected 95% confidence interval (CI)). The top/bottom values represent Japan/Italy. The 95% CI is estimated using a bootstrap method of 1000 samples. * P < 0.05; † P < 0.10. e: error term. chi-square (CMIN): < 0.001, comparative fit index: 0.887, standardized root mean squared residual: 0.069, root mean square error of approximation: 0.088, Akaike’s information criterion: 147.654.

Discussion

This study investigated the BJD and HRR in five countries (i.e., Japan, the US, the UK, Italy, and China), as well as their trends from 2020 to 2022. The associations between BJD and HRR in each country and each year were further investigated, and the direction of the associations between BJD and HRR was examined using a cross-lagged panel model in Japan and Italy. The results showed differences by country and year in BJD and HRR. The BJD increased from 2020 to 2021 in Japan, the US, the UK, and Italy.

Given the potential for BJD to promote discrimination and prejudice, it is important to understand the underlying reasons for the increase in BJD from 2020 to 2021. Furthermore, there were no large differences in BJD between 2021 and 2022, although the number of COVID-19 patients markedly increased from 2020 to 2021 and from 2021 to 2022 in these countries (Mathieu et al., 2020). We considered two possible interpretations. The first hypothesis is that the knowledge of infection control measures improved after the COVID-19 outbreak. For example, avoidance of the three Cs (i.e., closed spaces, close-contact settings, and crowded places), mask wearing, ventilation, and disinfection are known effective measures that have been promoted in many countries, resulting in increased adherence to infection control measures by 2020 (Petherick et al., 2021). Knowledge of these infection control measures may lead people to believe that infected individuals are infected because of the lack of adherence.

The second hypothesis is an increased threat related to COVID-19. Jackson et al. (2019) analyzed observational and intervention studies and reported that prejudice increases via increased tightness (i.e., strong norms and low tolerance of deviant behavior) in response to severe environments, including infectious diseases and natural disasters. Similar to the association between severe threats and tightness, it is also possible that the growing threat posed by the increasing number of individuals with COVID-19 increased the BJD, particularly between 2020 and 2021, before the wide availability of the vaccine.

In contrast, the HRR declined over time in Japan, the US, the UK, and Italy. This might be because the citizens in these countries regarded years of 2021 and 2022 as no longer “emergencies.” Alternatively, it might be that political control measures, such as lockdowns and staying at home, have led to the recognition of various disadvantages (e.g., slowed economic activity). “Pandemic fatigue” was reported to have occurred worldwide by 2020 owing to the prolonged COVID-19 pandemic (Petherick et al., 2021).

The differences in the BJD and HRR among the countries were interesting. The BJD was higher in Japan and lower in the UK. In contrast, the HRR was lower in Japan and higher in China. The trends in HRR were consistent with the strength of regulatory control measures in each country: requests for self-restraint and no legal restrictions were applied in Japan; there were lockdowns with penalties in the US, the UK, and Italy; and China had the strictest controls (Otsuyama et al., 2020). The strength of national regulatory control measures may reflect the HRR of each country, or vice versa. As noted above, there have been small changes in the BJD and HRR over time in China. This might reflect a refusal bias rather than the participants’ thoughts about BJD and HRR. Munro (2018) noted that refusal bias can influence responses to politically relevant questions in China. A similar refusal bias may have occurred in the present study.

Unlike those for HRR, cross-country differences in the BJD were challenging to interpret. Therefore, we attempted to examine the differences among countries concerning items similar to the BJD. First, comparable results were obtained for the BIJ, the item on which the BJD was based, at least between Japan and the U.S. (Murayama, Miura & Furutani, 2021). A previous study (Murayama, Miura & Furutani, 2021) measured BIJ assuming a hypothetical scenario where a character was injured in an accident. BIJ was found to be higher in Japan than in the US under a low moral condition (i.e., assuming that the character has a low moral): Cohen’s d was calculated as 0.92 from their arithmetic means and standard deviations. For the BJD in Japan and the US in this study, Cohen’s d was calculated from their arithmetic means and standard deviations to be 0.87 in 2020, 0.57 in 2021, and 0.75 in 2022. Second, a similar item that helps in interpreting the differences in BJD among countries observed in this study is tightness. Environmental threats increase tightness, eventually reinforcing prejudice, as reported by Jackson et al. (2019); among the four countries included in their study, tightness was the highest in Japan and lowest in the UK; Italy was excluded from the survey. Another study showed higher tightness in Japan than in the other four countries (Gelfand et al., 2011). Thus, it can be summarized that the differences in the BJD regarding COVID-19 among countries obtained in this study were similar to the differences in not only BIJ but also tightness among countries. Differences in BJD among countries may be partially related to this culturally shaped tightness in response to perceived threats from the environment, including past infectious disease outbreaks and natural disasters.

Positive associations were found between BJD and HRR in Japan and Italy. Interestingly, the strength of this association varied by country. However, a similar trend of positive association was also observed in the US in 2021 and 2022 and in the UK in 2021. Thus, it was possible that the associations between BJD and HRR were common in Japan, Italy, the US, and the UK, although the strength of the associations varied by country and year. As mentioned above, in Japan and Italy, although the BJD increased and the HRR decreased over time, the positive associations were stronger in 2022 than in 2020. This did not mean that participants had an unbiased decline in HRR and an increase in BJD in the population as a whole; rather, it meant that the HRR declined for those who had low BJD and relatively high HRR or that the BJD increased for those who had high HRR and relatively low BJD, or both. That is, either a weakening of HRR in those with low BJD, an increase in BJD in those with high HRR, or both might occur.

The cross-lagged panel model showed that bidirectional associations between BJD and HRR occurred between 2021 and 2022. That is, Japan and Italy experienced both a weakening of HRR in those with low BJD and an increase in BJD in those with high HRR. It should be noted that while the BJD in Japan and Italy did not increase from 2021 to 2022 (Table 2), the HRR in Italy decreased significantly (Table 3), and the HRR decreased for panel participants in both countries (Table S11). It is likely that the weakening of the HRR, especially among those with low BJD, between 2021 and 2022 led to a decrease in the HRR for the population as a whole.

Except for the BJD from 2021 to 2022 and the partial correlation between the BJD and HRR in 2022, there were no large differences in structure between Japan and Italy. The cross-lagged panel model included those who participated in all three surveys, and the results showed that changes in BJD and HRR over time were similar to those in the analysis of all participants. However, the trend for the association between BJD and HRR was different in Italy. There was a bias specific to participants who participated in all three surveys in Italy, which might explain the higher partial correlation between the BJD and HRR in 2022 (Fig. 1).

The following mechanism can be used to interpret the impact of BJD on the HRR. Those with a low BJD did not think that the infection was caused by the victim or that punishment should be imposed against the victim, resulting in disagreement with the rules enacted by the government. The influence of the HRR on the BJD can be interpreted as follows: those with high HRR strictly followed the rules and reinforced the belief that infected individuals must have broken the rules, leading to increased BJD.

Therefore, it may be beneficial to approach people with a high HRR during the early stages of an infectious disease outbreak to decrease the BJD, eventually reducing discrimination and prejudice. In addition to promoting infection control measures, it is important to disseminate public messages that infection is not necessarily the fault of the infected individuals. Such awareness-raising activities have been conducted since the COVID-19 outbreak (Ministry of Justice, 2022), but it is necessary to develop more effective approaches for individuals with a high HRR. The development of individualized messages effective for specific populations is promising. However, it should be noted that there are differences in BJD, HRR, and their associations between countries, as shown in this study. Approaches that are effective in Japan and Italy may not be as effective in other countries. Developing approaches to reduce discrimination and prejudice must be undertaken according to the characteristics of the country.

This study has several limitations. First, the online surveys conducted may have included a selection bias. Selection bias results from the choices made by the participants and investigators (i.e., researchers). Regarding the former, we attempted to reduce bias by providing rewards to participants. This had the advantage of encouraging participation even among those who were not interested in the research topic. Regarding the latter, we attempted to reduce the bias by adjusting for age, gender, academic career, and presence or absence of children and elderly people. Although efforts were made to reduce the effects of bias in this study, future studies should be conducted using alternative survey methods, such as postal and visiting methods, to further verify the external validity of the results. Secondly, this was an observational study. Although the cross-lagged panel model demonstrated the direction of the association between BJD and HRR, a causal relationship was not identified. Therefore, interventional studies aimed at decreasing the incidence of BJD are needed to verify the findings. Third, although this study revealed differences in BJD, HRR, and their associations across countries, it did not fully identify the underlying mechanism. This warrants further observational or interventional studies to investigate which factors, such as the cultural characteristics of each country and environmental threats (Jackson et al., 2019), govern BJD and HRR. Fourth, the study was conducted until the early stages of the omicron variant outbreak, when the number of infected individuals increased rapidly. Therefore, our study did not include how the subsequent deregulation of COVID-19 in each country changed the course of BJD and HRR. Further longitudinal surveys are required to determine the relationship between deregulation and BJD and HRR.

Conclusions

This study found that the BJD was higher in Japan and lower in the UK. The BJD increased significantly from 2020 to 2021 in all countries, except China. There were no significant changes in BJD between 2021 and 2022. Meanwhile, the HRR was lower in Japan and higher in China. The HRR decreased from 2020 to 2021 in Japan and from 2020 to 2022 in the US, the UK, and Italy. In Japan and Italy, there were significant positive associations between BJD and HRR. The cross-lagged panel model revealed positive bidirectional associations between BJD and HRR in Japan and Italy. This highlighted that HRR weakened for those with low BJD and BJD increased for those with high HRR. In Japan and Italy, it is promising to develop and disseminate effective public messages for those with a high HRR during the early stages of an infectious disease outbreak to decrease the BJD, eventually reducing discrimination and prejudice.

NOTES

A preliminary result was presented at the 61st conference of Japanese Society of Social Psychology (Miura et al., 2020)

Supplemental Information

Supplemental Information 1 Data of each respondent

Click here for additional data file.

Supplemental Information 2 Spearman-Brown coefficient for belief in just deserts by country and year

Click here for additional data file.

Supplemental Information 3 Cronbach’s α for human rights restriction by country and year

Click here for additional data file.

Supplemental Information 4 Belief in just deserts (BJD) by country and year. Data are shown as the mean (95% confidence interval)

Simple main effects are adjusted by Bonferroni correction: P values are multiplied by the number of groups (i.e., 5 for countries and 3 for years). BJD is adjusted for different covariates: age (33.2), gender (women = 0.60), academic career (university degree or higher = 0.64), children under junior high school age in the family (presence = 0.28), and elderly people over 65 in the family (presence = 0.22). Interaction: P = 0.002, partial η2 = 0.004.

Click here for additional data file.

Supplemental Information 5 Human rights restriction (HRR) by country and year

Data are shown as the mean (95% confidence interval). Simple main effects are adjusted by Bonferroni correction: P values are multiplied by the number of groups (i.e., 5 for countries and 3 for years). HRR is adjusted for different covariates: age (33.2), gender (women = 0.60), academic career (university degree or higher = 0.64), children under junior high school age in the family (presence = 0.28), and elderly people over 65 in the family (presence = 0.22). Interaction: P < 0.001, partial η2 = 0.023.

Click here for additional data file.

Supplemental Information 6 Belief in just deserts by country and year only for the first-time participants

Data are shown as the mean (95% confidence interval). Simple main effects are adjusted by Bonferroni correction: P values are multiplied by the number of groups (i.e., 5 for countries and 3 for years). Interaction: P ¡ 0.001, partial η2 = 0.010.

Click here for additional data file.

Supplemental Information 7 Human rights restriction by country and year only for the first-time participants

Data are shown as the mean (95% confidence interval). Simple main effects are adjusted by Bonferroni correction: P values are multiplied by the number of groups (i.e., 5 for countries and 3 for years). Interaction: P ¡ 0.001, partial η2 = 0.027.

Click here for additional data file.

Supplemental Information 8 Belief in just deserts (BJD) by country and year only for the first-time participants

Data are shown as the mean (95% confidence interval). Simple main effects are adjusted by Bonferroni correction: P values are multiplied by the number of groups (i.e., 5 for countries and 3 for years). BJD is adjusted for different covariates: age (31.5), gender (women = 0.60), academic career (university degree or higher = 0.65), children under junior high school age in the family (presence = 0.29), and elderly people over 65 in the family (presence = 0.23). Interaction: P ¡ 0.001, partial η2 = 0.007.

Click here for additional data file.

Supplemental Information 9 Human rights restriction (HRR) by country and year only for the first-time participants

Data are shown as the mean (95% confidence interval). Simple main effects are adjusted by Bonferroni correction: P values are multiplied by the number of groups (i.e., 5 for countries and 3 for years). HRR is adjusted for different covariates: age (31.5), gender (women = 0.60), academic career (university degree or higher = 0.65), children under junior high school age in the family (presence = 0.29), and elderly people over 65 in the family (presence = 0.23). Interaction: P ¡ 0.001, partial η2 = 0.027.

Click here for additional data file.

Supplemental Information 10 Partial correlation between belief in just deserts and human rights restriction by country and year

Controlled variables: age, gender, academic career, and presence or absence of children and elderly people. Bootstrap method with 1000 samples for each analysis is applied to estimate the 95% confidence interval (CI). P values are adjusted by Bonferroni-correction, that is, P values are multiplied by the number of stratified analyses (n = 15).

Click here for additional data file.

Supplemental Information 11 Belief in just deserts by Japan/Italy and by year

Data are shown as the mean (95% confidence interval). Interaction: P = 0.321, partial η2 = 0.008. Min effect: year, P = 0.289, partial η2 = 0.008; country, P < 0.001, partial η2 = 0.131.

Click here for additional data file.

Supplemental Information 12 Human rights restriction by Japan/Italy and by year

Data are shown as the mean (95% confidence interval). Interaction: P = 0.084, partial η2 = 0.017. Main effect: year, P < 0.001, partial η2 = 0.183; country, P = 0.034, partial η2 = 0.030.

Click here for additional data file.

Supplemental Information 13 Pearson correlation between belief in just deserts and human rights restriction by country and year

P values are adjusted by Bonferroni correction, that is, P values are multiplied by the number of stratified analyses (n = 6). CI: confidence interval.

Click here for additional data file.

We would like to thank Editage for English language editing.

Additional Information and Declarations

Competing Interests

Author Contributions

Human Ethics

Data Availability

The authors declare there are no competing interests.

Michio Murakami conceived and designed the experiments, analyzed the data, prepared figures and/or tables, authored or reviewed drafts of the article, and approved the final draft.

Kai Hiraishi conceived and designed the experiments, performed the experiments, authored or reviewed drafts of the article, and approved the final draft.

Mei Yamagata conceived and designed the experiments, authored or reviewed drafts of the article, and approved the final draft.

Daisuke Nakanishi conceived and designed the experiments, performed the experiments, authored or reviewed drafts of the article, and approved the final draft.

Andrea Ortolani conceived and designed the experiments, performed the experiments, authored or reviewed drafts of the article, and approved the final draft.

Nobuhiro Mifune conceived and designed the experiments, performed the experiments, authored or reviewed drafts of the article, and approved the final draft.

Yang Li conceived and designed the experiments, performed the experiments, authored or reviewed drafts of the article, and approved the final draft.

Asako Miura conceived and designed the experiments, performed the experiments, authored or reviewed drafts of the article, and approved the final draft.

The following information was supplied relating to ethical approvals (i.e., approving body and any reference numbers):

This study was approved by the Osaka University Graduate School of Human Sciences Research Ethics Committee.

The following information was supplied regarding data availability:

The raw data are available in the Supplementary File.

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
