# Peer review of "Differences in and associations between belief in just deserts and human rights restrictions over a 3-year period in five countries during the COVID-19 pandemic"

_PeerJ, doi:10.7717/peerj.16147_

## Round 0.1 · original submission · Major Revisions

The primary issue I have with the manuscript lies in the between-country and annual comparisons it presents. There is a high probability that the differences identified in BJD and HRR between countries and between years can be attributed to differences in age and gender, as well as other unreported sociodemographic variables. While the authors employed a MANOVA to account for potential confounding factors such as age and gender, this approach may not sufficiently address the issue. Would it be possible for the authors to incorporate methods like propensity score matching to further substantiate their findings? This might present a more robust justification and thereby strengthen the overall validity of their research.

·

Basic reporting

The manuscript is based on impressive empirical evidence and makes an original contribution. Only minor revisions are needed before it can be published.
1. Their explanation of the relationship between discrimination and BJD seems somewhat inconsistent and confusing. In early chapter, they emphasized BJD as a major factor promoting discrimination (66-71), but in the latter chapter, BJD was connected with the lack of adherence to infection control measures and tightness(282-301). In the last, HRR is emphasized as a major source of discrimination and prejudice(389-392).
2. In relation to the above issue, I can’t understand well the difficulty of interpreting BJD(312-314). I believe that they should describe this topic in detail.

Experimental design

1. The large gaps of BJD between in Japan and in the other countries lead us to to be suspicious of the equivalence of the questions regarding BJD or BIJ. Is the gap of BIJ as large as that of BJD?
In Japanese "Jigoujitoku" may include various aspects of causality, such as the lack of adherence to infection control measures. They need more explanation on this issue.
2. Their argumentation are based on the strong relationship of BIJ and BJD. However, BIJ itself was not measured in this survey. If they apply the findings of previous studies of BIJ to their BJD study, they should offer some evidence, or some explanation at least.

Validity of the findings

no comment

Additional comments

469 Norman F. Cantor should be changed into Canter, NF

·

Basic reporting

The text appears to be written in professional, plain, and unambiguous English. It discusses the context and history of discrimination and prejudice against infectious diseases, citing pertinent research to support its claims. The references are properly cited, and the formal tone is maintained throughout.

The text introduces the global public health issues of discrimination and prejudice against infectious diseases and emphasizes historical examples of such discrimination, such as the discrimination of Jews during the pandemic of the plague. The section then provides examples of recent discrimination against people infected with diseases such as acquired immunodeficiency syndrome, the Zika virus, leprosy, and COVID-19.

The text discusses the belief in just deserts (BJD), a psychological factor that may contribute to discrimination and prejudice, and its association with belief in immanent justice (BIJ) and belief in a just world. To substantiate these claims, pertinent studies and citations are provided.

The text also acknowledges the heterogeneity of sociopsychological factors associated with infectious diseases across countries, as well as the ambiguity surrounding the relationship between BJD and human rights restrictions (HRR) in countries other than Japan or after 2020.

Examining the differences in BJD and HRR across countries and years, exploring the associations between BJD and HRR in each country and year, and examining the direction of the associations in countries where positive associations between BJD and HRR were identified are the stated objectives of the study.

Overall, the text is well-written and effectively conveys the study's context, background, and goals. It is suitable for a research paper or scholastic publication due to its professional and academic tone.

Experimental design

The research paper emphasises on examining the beliefs and attitudes of individuals in various countries regarding COVID-19. The study's methodologies are described in sufficient detail to permit comprehension of the data collection and analysis procedure. The main aspects of the research determine whether or not it is original primary research within the journal's scope and demonstrates a high level of technical and ethical rigour.

The research question appears to be well-defined and pertinent as it investigates the differences in beliefs and attitudes towards COVID-19 among participants from various countries. The purpose of this query is to examine the relationship between beliefs regarding infection responsibility and acceptance of government control measures.

Knowledge Gap: The paper mentions that COVID-19 control measures have been implemented in numerous countries, but that the policies, penalties, and legal bases for these measures have varied. The purpose of this research is to close this knowledge deficit by investigating the relationship between these differences in control measures and individuals' beliefs regarding COVID-19 infection responsibility and acceptance of government control measures.

The study is characterised as a longitudinal online survey, implying that data were gathered from participants in multiple phases over the course of several years. This design permits the examination of trends and shifts in attitudes and beliefs over time.

The participants were recruited from five countries (Japan, the United States, the United Kingdom, Italy, and China) and were compensated for their participation. The sample size appears adequate (approximately 400 valid samples per survey in each country) for each country.

The Osaka University Graduate School of Human Sciences Research Ethics Committee approved the study, and participant consent was obtained prior to administering the questionnaire.

The data was collected by means of an online survey administered at various times for each country. There were questionnaires available in Japanese, English, Italian, and Chinese to facilitate participants from various regions.

The authors employed a variety of statistical analyses, including multivariate analysis of variance (MANOVA) and correlation analysis, to investigate the relationships between variables. To maintain the statistical rigour of their findings, they also implemented the proper corrections for multiple comparisons.

According to the information provided in the methods section, the research appears to be an original primary study that falls within the scope of the journal. The research query is pertinent and significant, and the study design and methodology are technically and ethically rigorous. Multiple countries and a longitudinal approach contribute to the reliability of the findings. However, a comprehensive evaluation of the paper would necessitate a comprehensive examination of the entire manuscript, including the results and discussion sections.

Validity of the findings

The study's findings appear to be valid, and the authors have made an effort to interpret and discuss them exhaustively. Over the course of three years, they investigated beliefs about COVID-19 infection responsibility (BJD) and acceptance of government control measures (HRR) in five countries. The study employed a longitudinal study design and statistical analyses to investigate the associations between BJD and HRR, as well as the trends in these beliefs over time.

---

## Round 0.2 · accepted · Accept

Thank you for addressing the editor and reviewers' concerns.